# Insect Body Defence Reactions against Bee Venom: Do Adipokinetic Hormones Play a Role?

**DOI:** 10.3390/toxins14010011

**Published:** 2021-12-23

**Authors:** Karolina Bodláková, Jan Černý, Helena Štěrbová, Roman Guráň, Ondřej Zítka, Dalibor Kodrík

**Affiliations:** 1Biology Centre, Institute of Entomology, CAS, Branišovská 31, 370 05 Ceske Budejovice, Czech Republic; Bodhys14@seznam.cz (K.B.); Honya.c@seznam.cz (J.Č.); hradova@email.cz (H.Š.); 2Faculty of Science, University of South Bohemia, Branišovská 31a, 370 05 Ceske Budejovice, Czech Republic; 3Department of Chemistry and Biochemistry, Faculty of AgriSciences, Mendel University in Brno, Zemědělská 1665/1, 613 00 Brno, Czech Republic; R.Guran@email.cz (R.G.); zitkaondra@gmail.com (O.Z.); 4Central European Institute of Technology, Brno University of Technology, Purkyňova 656/123, 612 00 Brno, Czech Republic

**Keywords:** adipokinetic hormone, American cockroach, dopamine, honey bee, melittin, metabolism, muscle ultrastructure, vitellogenin

## Abstract

Bees originally developed their stinging apparatus and venom against members of their own species from other hives or against predatory insects. Nevertheless, the biological and biochemical response of arthropods to bee venom is not well studied. Thus, in this study, the physiological responses of a model insect species (American cockroach, *Periplaneta americana*) to honeybee venom were investigated. Bee venom toxins elicited severe stress (LD_50_ = 1.063 uL venom) resulting in a significant increase in adipokinetic hormones (AKHs) in the cockroach central nervous system and haemolymph. Venom treatment induced a large destruction of muscle cell ultrastructure, especially myofibrils and sarcomeres. Interestingly, co-application of venom with cockroach Peram-CAH-II AKH eliminated this effect. Envenomation modulated the levels of carbohydrates, lipids, and proteins in the haemolymph and the activity of digestive amylases, lipases, and proteases in the midgut. Bee venom significantly reduced vitellogenin levels in females. Dopamine and glutathione (GSH and GSSG) insignificantly increased after venom treatment. However, dopamine levels significantly increased after Peram-CAH-II application and after co-application with bee venom, while GSH and GSSG levels immediately increased after co-application. The results suggest a general reaction of the cockroach body to bee venom and at least a partial involvement of AKHs.

## 1. Introduction

Honeybee venom is a cocktail of various biologically active substances including peptides, proteins, amines, and amino acids dissolved in an aqueous solution [1]. Toxins either trigger biochemical cascades or become biologically active following the effects of another toxin through positive feedback loops. Bee venom mainly contains melittin (approximately 50% of dry matter), phospholipases, hyaluronidase, other enzymes, and small, biologically active substances with free amine groups (dopamine, histamine, serotonin, and norepinephrine) [1,2]. Melittin is a 26-amino acid peptide with numerous biological, toxicological, and pharmacological activities [3,4]. Toxicity is based on an ability to bind to the phospholipid bilayer of cell membranes and form pores. Melittin also activates other enzymes, including phospholipases, which break down phospholipids and increase cell destruction [1,4], which is the likely cause of mitochondrial damage and collapse [5]. Biogenic amines also play important roles as major stimulators of inflammation and other immune responses [2,6]. The defence reaction against bee venom in the victim’s body is generally well understood; however, not all the details are known. The recent identification of vitellogenin in bee venom [7] suggests that it increases the allergenic properties of the venom; however, this has not been studied in detail. Vitellogenins are mostly known as yolk precursors involved in reproduction, but they also play important roles in protection against oxidative stress, wound healing, and insect immunity, with strong activity against bacteria and other pathogens [8,9].

Bee venom intoxication certainly elicits severe stress on the affected body, with the anti-stress response generally controlled by the nervous and endocrine systems [10]. In insects, anti-stress reactions are predominantly regulated by adipokinetic hormones (AKHs), which are responsible for maintaining homeostasis. AKHs are small peptides (8–10 amino acids in length) that are released from the corpora cardiaca, a small neuroendocrine gland connected to the brain [11,12]. They interact with various body systems involved in the activation of metabolism and related processes (reviewed in [11,12,13]), which include activation of digestive enzymes [14,15,16], mobilisation of anti-oxidative stress response (reviewed in [13]), and/or interactions of AKHs with biogenic amines [17,18], among others. It is also known that some biogenic amines can control AKH production [17]. Nonetheless, it remains unclear whether these mechanisms are involved in AKH responses induced by natural venoms.

In this study, the model American cockroach, *Periplaneta americana*, was utilised to measure the physiological effects of bee toxins. This is a very popular model for physiological and biochemical studies and is predominantly used by American insect scientists. The hormonal system of this species is well known and possesses two AKHs called Peram-CAH-I (*P. americana* cardioaccelerating hormone I (pGlu-Leu-Thr-Phe-Thr- Pro-Asn-Trp-NH_2_); and Peram-CAH-II (pGlu-Val-Asn-Phe-Ser-Pro-Asn-Trp-NH_2_) [19]. Cockroaches are not the primary targets of bee attacks; however, the main toxic activities of bee venom by molecules such as melittin or phospholipases are not species-specific. Bee venom should be effective against cockroaches since bees often face attacks by other insect species. The main goals of this study were to describe the effect of bee toxins on several metabolic characteristics and on muscle ultrastructure. Furthermore, the possible role of AKHs and several other biologically active substances (vitellogenins, dopamine, and glutathione) in defence reactions against the venom was evaluated.

## 2. Results

### 2.1. General Interactions between Venom and AKH

Initial experiments involved measuring cockroach mortality 24 h after applying increasing doses of bee venom to determine the appropriate dose (Figure 1). A dose of 0.5 uL venom per cockroach was deemed the most appropriate since it elicited approximately 20% mortality (LD_50_ = 1.063 uL) and provided a sufficient physiological effect whilst leaving a satisfactory number of living insects for the following experiments. The estimated half-life of 3.82 min for melittin was used to assess the rate at which bee venom is degraded by the cockroach defence system in the haemolymph (Figure 2).

Venom treatment elicited severe stress in the cockroach body, since AKH levels in the central nervous system (CNS) significantly rose by 3.8-fold and 3.3-fold at 6 h and 12 h, respectively (Figure 3). The levels returned to the control level after 24 h. Corresponding AKH levels in the haemolymph significantly increased by 3.6-fold and 1.4-fold after 1 h and 2 h, respectively (Figure 4), indicating that AKH turnover in the haemolymph was much faster than that in the CNS. Interestingly, the level of AKHs in the CNS was in the order of pmols, while in haemolymph in the order of fmols.

### 2.2. Effect of Venom and AKH on Muscle Ultrastructure

Envenomation changed the muscle ultrastructure in the cockroach thorax (Figure 5) by inducing myofibril destruction and general disorganisation of organelles in muscle cells (Figure 5C). Interestingly, co-application with Peram-CAH-II prevented this damage (Figure 5D). Meanwhile, hormone administration alone (Figure 5B) did not affect the common ultrastructure of muscle cells compared with the Ringer saline control (Figure 5A).

### 2.3. Effect of Venom and AKH on Selected Metabolites

The levels of the main haemolymph nutrients in the cockroach (carbohydrates, lipids, and proteins) were modulated by the venom treatment after 6 h (Figure 6), while the changes were less significant with increasing time (data not shown). Carbohydrate levels significantly decreased after envenomation (1.8-fold) and significantly increased after Peram-CAH-II treatment (1.8-fold; Figure 6A) as expected, since this hormone primarily mobilises carbohydrates in the cockroach body [20]. Co-treatment with both agents eliminated these fluctuations and maintained carbohydrate levels at that of the control. The lipid levels significantly increased in all experimental groups after the treatments (Figure 6B); however, these changes were slightly smaller than those recorded for carbohydrates (1.3–1.7-fold), with the total lipid level in the haemolymph approximately two times lower than the total carbohydrate level. Protein levels significantly increased by 1.2–1.3-fold following Peram-CAH-II treatment alone or in combination with bee venom (Figure 6C).

The activity of the main digestive enzymes in the cockroach midgut after bee venom application showed some significant changes, but these were generally small and ambiguous. Therefore, only the temporal effects of the venom, Peram-CAH-II, and their co-addition are shown (Figure 7). Bee venom slightly stimulated amylase activity after 6–24 h, which was not modulated by Peram-CAH-II. The 2-fold stimulation using Peram-CAH-II alone was not observed until 24 h after treatment (Figure 7A). Lipase activity significantly increased by 2-fold 24 h after Peram-CAH-II treatment, while other treatments had insignificant effects (Figure 7B). Protease activity was significantly stimulated 12 h and 24 h after venom administration, while Peram-CAH-II alone only stimulated protease activity by 2.2-fold after 24 h. Interestingly, Peram-CAH-II stimulation of all enzymes (24 h after treatment) was significantly eliminated by the venom during co-application of both agents (one-way ANOVA with Tukey’s post-test: amylase, *p* = 0.0004; lipase, *p* = 0.0015; protease, *p* = 0.002; data not shown for clarity).

Vitellogenins are actively involved in defence reactions against various pathogens [9]. There was a significant reduction in both ~100 kDa vitellogenin proteins by 8.2-fold and 2.9-fold in the female haemolymph 6 h after venom administration (Figure 8A,B). Meanwhile, the response in the male haemolymph was less pronounced, with the heavier protein significantly reduced by 1.3-fold and the lighter protein unaffected (Figure 8A,C).

Bee venom application insignificantly stimulated dopamine levels in the cockroach CNS 6 h after treatment (Figure 9). Nevertheless, a significant increase in this amine was recorded after treatment with Peram-CAH-II alone or together with venom by 2.3-fold and 2.4-fold, respectively. Meanwhile, the levels of glutathione oxidative stress markers (reduced and oxidised glutathione: GSH and GSSG, respectively) slightly increased following treatment with venom alone or hormone alone (Figure 10). However, their co-administration led to a significant increase in GSH and GSSG levels by 5.3-fold and 2.8-fold, respectively.

## 3. Discussion

Bees probably developed from ancient wasps approximately 120–130 million years ago [21,22]. However, it cannot be excluded that its sting apparatus may have contributed to the successful evolution of bees, owing its highly effective mechanical and chemical protective effect. It is well known in beekeeping practice that the venom, possible owing to its non-specific mechanism, works well against both mammals, including humans, and insects.

### 3.1. Interactions between Bee Venom and AKH

The half-life of the main component of bee venom (melittin) was approximately 4 min in the cockroach haemolymph (see Figure 2). This is the first estimate of the degradation rate of this toxin in insect or vertebrate bodies, to the best of our knowledge.

We hypothesise that the phospholipase activity of melittin itself is limited to the first steps of envenomation, while the activity of melittin-stimulated phospholipases, which are responsible for the major cell disruption, is manifested later on. Melittin belongs to relatively small peptides that are readily degraded in the insect haemolymph. The half-lives of some adipokinetic peptides (8–10 amino acids long) were similarly in the order of a few minutes. For example, Locmi-AKH-I, -II, and -III from flying locusts (*Locusta migratoria*) have half-lives of 35, 37, and 3 min, respectively [23,24], while the half-life of Pyrap-AKH in resting firebugs (*Pyrrhocoris apterus*) is approximately 18 min [25].

Stress induced by the application of bee venom caused a significant increase in AKH levels in the cockroach CNS and haemolymph (see Figure 3 and Figure 4), with AKH levels fluctuating much more quickly in the haemolymph than in the CNS. This is unsurprising because there is only a loose coupling between the biosynthesis and release of AKH [17]. Thus, there was probably some inertia in the synthesis of AKH in the cockroach CNS after venom application. Higher levels of this stress hormone are thought to provide faster and more effective mobilisation of the defence system. An increase in AKH levels in the insect body is reported after intoxication with synthetic or natural toxins, including insecticides [26,27,28] and toxins produced by entomopathogenic nematodes [29] and fungi [30,31].

However, the immediate elevation of AKH levels in the insect body after toxin treatment is paradoxically counterproductive because it increases insect mortality. This is presumably caused by the stimulation of metabolism by AKH, observed by increased carbon dioxide production, which intensifies the action of toxins by accelerating their penetration into active tissues and cells before the defence system is fully activated [29,31,32].

Bee venom also causes myonecrosis [33,34], although the phenomenon is not studied very intensively. Melittin binds to a specific binding site on the myosin light chain [35]. The application of melittin and venom phospholipase results in disorganised and disrupted myofibrils of mouse skeletal muscles 24 h after treatment [36]. A similar effect was observed when the both toxins were applied to human skeletal muscle cell cultures [37]. Therefore, it is unsurprising that bee venom application caused extensive destruction of cockroach muscle cell ultrastructure in this study (see Figure 5); the myofibril fibres disintegrated, the sarcomeres were damaged, and the whole muscle cells were disorganised. However, co-application of Peram-CAH-II with the venom prevented extensive damage and maintained a muscle cell ultrastructure similar to that of the control. Further studies are required to show if AKHs have biological activity against other toxins. Although the mechanism of action is unknown, the effect may be indirectly mediated by the stimulatory effect of AKH on the immune system [38,39].

### 3.2. Effect of Venom/AKH on Physiological Processes

*P. americana* primarily use carbohydrate metabolism controlled by Peram-CAH-I and -II [19,40]. In this study, carbohydrate levels were approximately double those of lipids in the cockroach haemolymph, followed by a significant increase in carbohydrate levels after Peram-CAH-II treatment (see Figure 6). Carbohydrate levels significantly decreased after bee venom application; however, the co-application of venom and Peram-CAH-II resulted in similar levels to the control. The same pattern was recorded when cockroaches were treated with the entomopathogenic fungus, *Isaria fumosorosea* [30]. It is possible that carbohydrates are used to energetically cover defence processes against toxins. Haemolymph lipid levels were slightly elevated after application of both agents separately; however, their co-application did not lead to an accumulation of the effect. Mobilised lipids may supplement the energy role of carbohydrates in general defence reactions. Protein levels were slightly affected by both agents, without an apparent role. Venom application slightly stimulated amylase and protease activity, with no stimulation of lipases in the midgut. This may reflect non-specific effects associated with a general increase in metabolic stress. Peram-CAH-II does not modulate the effect of the venom despite the stimulatory role of AKHs on gut function [14,15]. Stimulation of digestive enzymes by AKH is probably the result of a multistage cascade, as the effect is not visible until 24 h [16,41], which was corroborated by our study. Elimination of the stimulatory effect of Peram-CAH-II by venom co-application after 24 h similarly implies that a cascade of reactions is involved.

Vitellogenins (egg yolk precursor proteins) are primarily present in egg-laying females but are also present in males from some insect species because their actions are broader than their name implies. They are involved in defence reactions against oxidative stress, various infections, pathogens, and their toxins [42,43,44]. This study concentrated on the levels of the two most abundant types of vitellogenin in *P. americana* [45], observed at ~100 kDa (Figure 8). Vitellogenin levels significantly decreased after venom treatment in the female haemolymph, comparable with levels in the haemolymph of *P. apterus* females when treated with nematodal and fungal infections [9]. Toxin-induced stress is thought to reduce vitellogenin levels in females so that they remain sufficient for defence but not for reproduction, which saves energy and helps females withstand intoxication. The response in males is more complicated, with the nematodal infection of *P. apterus* eliciting a significant increase in vitellogenin transcription and protein levels, while a fungal infection elicits increased transcription but does not increase protein levels [9]. In this study, a mild reduction of vitellogenin levels was recorded in males after bee venom application. Together, the data suggest that the vitellogenin response in males is unclear; it is probably species- and toxin-specific and might depend on the general level of vitellogenins in the male body. The effect of Peram-CAH-II on this anti-venom reaction was not studied due to these ambiguities. The application of Peram-CAH-II would further complicate the already unclear situation, since only small doses of AKH reduce vitellogenin synthesis [46,47].

Dopamine is a biogenic amine that plays numerous roles as a regulator of cell function. In insects, dopamine is mainly present in all parts of the nervous system [48] where it has neuromodulatory functions. Application of the neonicotinoid insecticide imidacloprid to *Drosophila melanogaster* results in decreased dopamine levels in the brain [49]. Nevertheless, no significant changes in dopamine levels were recorded in the cockroach brain after bee venom application in this study (see Figure 9). Interestingly, treatment of the same cockroach species by the venom of the parasitoid wasp *Ampulex compressa* results in behavioural changes in grooming activities probably mediated by the dopaminergic receptor [50]. Dopamine levels in cockroaches are stimulated by Peram-CAH-II (see Figure 9). There may be a feedback mechanism between dopamine and AKH in insects because dopamine treatment of the *P. apterus* body leads to a reduction in AKH levels in the haemolymph; however, dopamine levels in the CNS remain unchanged [18].

Surprisingly, the application of bee venom had no effect on the level of oxidative stress in the body of cockroaches (see Figure 10), determined by monitoring the levels of glutathione (GSH and GSSG). This is in contrast with previous examples showing that oxidative stress in the insect body increases with the application of insecticides/toxins, such as herbicide paraquat [51], and insecticides endosulfan and malathion in *P. apterus* [27], and *Galanthus nivalis* agglutinin and *Bacillus thuringiensis* toxin in *Leptinotarsa decemlineata* (Colorado potato beetle) [52]. AKH co-application with the toxins results in a significant reduction in oxidative stress in all of these examples. This study found that GSH and GSSG levels increased only when the venom and Peram-CAH-II were co-administered to the cockroaches. This may be explained by the mode of AKH action: AKH activity is often (but not always) realised when a stressor is present; for example, AKH injection into *L. migratoria* enhances the activity of phenoloxidase only when stressors (bacterial lipopolysaccharide or laminarin) are present, and no effect is observed when only AKH is applied [38]; for further examples see interactions between AKH and insecticides [13,28].

## 4. Conclusions

We have found that bee venom has significant effects on the physiological processes in the body of *P. americana* cockroaches; however, the main novelty of this study was that some effects are modulated by Peram-CAH-II. Envenomation elicits an increase in AKH levels in the cockroach CNS and haemolymph, and induces a large destruction of muscle cell ultrastructure. This latter effect is eliminated by Peram-CAH-II and venom co-application. Thus, AKH can be considered as an inhibiting factor of myonecrosis, which may have interesting theoretical and practical applications in the future. The venom also modulates the level of nutrients in cockroach haemolymph and digestive enzyme activity in the midgut, and reduces the level of vitellogenins, especially in females. Bee venom did not cause a significant increase in dopamine and glutathione (GSH and GSSG) levels; however, significant stimulation with Peram-CAH-II was observed, which will be further investigated.

## 5. Materials and Methods

### 5.1. Experimental Animals

A stock culture of the American cockroach, *Periplaneta americana* was used in the present study for details see [16,41]. Briefly, the colony was kept in 60 L glass fish tanks in a mass cultures and reared at constant temperature of 30 ± 1 °C under short-day conditions (12/12 h light/dark cycle). The cockroaches were fed dog food (VitalBite) and water *ad libitum*. To avoid possible complications with the female ovarian cycle, only males of unknown age were used in most of the experiments.

In vitellogenin experiments (see Section 5.8) 7-day old cockroach males and females were used. To prepare them grown nymphs were moved from the main colony to a 5 L glass, ecdysed adults were removed every day, and males and females were maintained separated until used.

Worker foragers of the honeybee *Apis mellifera carnica*, which served as the venom donors were maintained in the apiary of the Biology Centre in České Budějovice (Czech Republic; 48°58′31.924″ N, 14°26′44.671″ E; altitude 390 m). The bees were maintained according to standard beekeeping techniques [53].

### 5.2. Venom and Hormone Treatments

Experimental cockroaches were injected with crude bee venom, which was obtained from venom sacs dissected from the honeybee worker foragers. Briefly, dissected sacs were carefully torn with tweezers in a microtube and gently centrifuged (to avoid contamination by content of the venom gland cells), and the released venom was transferred into new microtubes (one worker sac contains approximately 2 uL of venom) that were kept at −80 °C until used. To avoid possible fluctuation on the amount of toxins in the crude venom [54], one pool of venom was used for all experiments. This venom was obtained from June bees (for the list of plants blooming in the vicinity of the apiary in June see Appendix A).

To determine the suitable venom dose to be used, mortality was assessed 24 h after application of increasing venom doses (0.25–4.0 uL). Three groups of cockroaches (*n* = 15–20) were used for test each dose.

In some experiments the cockroaches were injected with 40 pmol of adipokinetic hormone Peram-CAH-II (Polypeptide Laboratories, Praha, Czech Republic) [16] resuspended in 2 uL 20% methanol in Ringer saline. For the dose selection and other details see [16,41].

Four cockroach groups were used in the experiments evaluating the combined effects of bee venom and Peram-CAH-II: (1) cockroaches treated with Ringer saline (control), (2) with venom, (3) with Peram-CAH-II, (4) and with venom plus Peram-CAH-II. In most experiments, data was recorded at 6 h, 12 h, and/or 24 h after treatment depending on the nature of the marker.

### 5.3. Turnover of Melittin in the Cockroach Body

Using a Hamilton syringe, the cockroaches were injected in the thorax with 2 μL of crude venom with known content of melittin. Samples of haemolymph were collected over a period of 30 min from the insects; a drop of haemolymph exuding from the soft part of the 3rd leg base pricked with a pin was pipetted into microtubes, and 5 μL of this was taken up for the melittin determination. To calculate the half-life of the melittin turnover, the haemolymph volume in the cockroach body was measured. Using azorubin dye according to Smith [55] the haemolymph content was estimated to be about 200 μL.

Melittin level in haemolymph was estimated by reversed phase (RP)-high performance liquid chromatography (HPLC). Haemolymph samples were extracted in 80% methanol and after centrifugation the supernatants were evaporated in a vacuum centrifuge. Then, the dry pellets were resuspended in A:B solution (9:1 ratio; A: 0.11% trifluoroacetic acid in water; B: 0.1% trifluoroacetic acid in 60% acetonitrile). The samples were analysed on a HPLC system (Waters, Milford, MA, USA) operated using the Clarity software (version 8.0; DataApex, Prague, Czech Republic) with a fluorescence detector Waters 2475 (λ_Exitation_: 280 nm; λ_Emission_: 348 nm) using a Chromolith Performance RP-18e column 150–4.6 mm (Merck Life Science, Darmstadt, Germany) with a flow rate of 1 mL/min and the following gradient: 0–2 min of 10% B, 2–5 min of 10–70% B, 5–16 min of 70–100% B, and 16–20 min of 100% B. Retention time of melittin under the used conditions was of 10.3 min. Melittin titre in the haemolymph samples was estimated using area of the melittin HPLC peak compared with that of the synthetic standard (Sigma-Aldrich, St. Louis, MO, USA).

### 5.4. Quantification of Peram-CAH-I and Peram-CAH-II in Cockroach Body

The brains along with the corpora cardiaca were dissected from the cockroach heads under the Ringer saline. Then, AKHs were extracted from the tissues using 80% methanol. The solutions were dried in a vacuum centrifuge and the resulting pellets were stored at −20 °C until further analysis. The level of AKHs in these samples was evaluated by the competitive enzyme-linked immunosorbent assay (ELISA) as previously described [56]. The primary antibody (dilution 1:2000; Sigma Genosys, Cambridge, UK) used in this assay recognises well both *P. americana* AKHs—Peram-CAH-I and Peram-CAH-II [25]. Know amount of synthetic Peram-CAH-II served as standard for calculating of AKH content in the experimental samples.

AKH levels were also evaluated in the cockroach haemolymph by RP HPLC. As previously described [25], haemolymph samples were pre-purified by solid phase extraction and 100 uL equivalent of these samples was analysed under the same HPLC conditions as described in Section 5.3. The following gradient protocol was used: 0–2 min of 10% B, 2–12 min of 10–90% B, and 12–15 min of 100% B. AKH levels in haemolymph were estimated based on the areas of the AKH HPLC peaks compared with those of synthetic standards, which had retention times of 8.79 and 10.03 min for Peram-CAH-I and Peram-CAH-II (both from Polypeptide laboratories), respectively. AKH losses during HPLC purification were of approximately 25% [25], for which a correction was included into the calculations.

### 5.5. Transmission Electron Microscopy (TEM) of Cockroach Muscle

*P. americana* thoracic muscles were dissected in Ringer saline and immediately fixed in 2.5% glutaraldehyde (electron microscopy grade) in 0.2 M phosphate buffer at 4 °C for 7 days. After fixation, the muscles were washed three times for 15 min with 0.1 M phosphate buffer (pH 7.4). Then, the samples were post-fixed in 2% osmium tetroxide in 0.1 M phosphate buffer for 2 h for better ultrastructure preservation and contrast. The samples were washed again, dehydrated in a series of acetone solutions, and embedded into Epon-Araldite resin. Ultrathin sections were cut using an ultramicrotome (Leica Biosystems, Wetzlar, Germany) with diamond knives (Diatome Ltd., Nidau, Switzerland) and examined under a JEOL JEM-1010 transmission electron microscope (JEOL, Tokyo, Japan). Images were captured using a Sis MegaView III CCD camera (EMSIS, Münster, Germany).

### 5.6. Quantification of Nutrients in Haemolymph

The levels of proteins, carbohydrates, and lipids were determined in 1 uL *P. americana* haemolymph samples after removing the haemocytes by centrifugation.

Protein quantification was performed using the Bicinchoninic Acid Protein Assay Kit (Sigma-Aldrich, St. Louis, MO, USA) [57]. Bovine serum albumin was used to establish a standard curve for protein quantification.

Lipid quantification was performed using the sulpho-phosho-vanillin method according to Zöllner and Kirsch [58], and as modified by Kodrík et al. [59]. Oleic acid was used to establish a standard curve for quantifying the lipids.

Free carbohydrate were assessed by the anthrone method [60], and as modified by Socha et al. [61]. Glucose was used to establish a standard curve for carbohydrate quantification.

### 5.7. Activity of Gut Enzymes

The activity of amylases, lipases and proteases was determined in the cockroach midgut. The midguts were homogenised (sonicated) in 0.1 M phosphate buffer pH 5.7 for amylase and/or 0.2 M tris pH 7.8 for lipase and protease assessment. The samples were then centrifuged and aliquots of 0.005 midgut equivalent were analysed as previously described [41].

The amylase assay was performed with DNS (3,5-dinitrosalicylic acid) reagent according to Bernfeld et al. [62], and as modified by Kodrík et al. [14]. Enzyme activity was calculated in μmol maltose/mg of fresh midgut weight.

Lipase activity was assessed using 4-methylumbelliferyl butyrate (4-MU) according to Roberts et al. [63], and as modified by Kodrík et al. [14]. Activity was expressed in nmol of 4-MU/min/mg of fresh midgut weight.

Protease activity was assessed with the resorufin-casein kit (Roche, Basel, Switzerland) according to the manufacturer’s instructions. The final activity was expressed in proteolytic activity units (U)/mg of fresh midgut weight. U was defined as the amount of enzyme (mg) which caused an increase in optical density by 0.1 per min in 1 mL of the reaction mixture [64].

### 5.8. Vitellogenin Level

Level of vitellogenin was quantified in the haemolymph of female and male cockroaches. The male and female samples were diluted by 10-fold and 25-fold in a sample buffer, respectively, and 10 μL each dilution was analysed by sodium dodecyl sulphate–polyacrylamide gel electrophoresis (SDS-PAGE) using commercial gels (Miniprotean TGX Gel, 4–20%, Bio-Rad, Hercules, CA, USA) according to Laemmli et al. [65], and as modified by Socha et al. [66]. The separated proteins in the gel were stained with Coomassie Brilliant Blue R-250, and the Vg bands were identified according to the molecular weight standards (10–250 kDa, Thermo Fisher Scientific, Waltham, MA, USA) [45]. Specific protein bands were evaluated using the GS-800 Calibrated Densitometer in Quantity One (Version 4.6) software (Bio-Rad, Hercules, CA, USA).

### 5.9. Dopamine and Glutathione Determination

Cockroach brains were dissected as described in Section 5.4, were sonicated in distilled water (three brains/sample), lyophilised, and stored in refrigerator until further analysis. The lyophilised samples were then diluted in 1 mL of water (liquid chromatography-mass spectrometry [LC-MS] quality) and were vortexed (30 s) and centrifuged (10 min at 23,500× *g*, 4 °C). Supernatant was passed through a 0.2-µm filter and placed in HPLC microvials with glass inserts (250 µL). HPLC-MS analysis was performed using a KNAUER PLATINBlue V6900A HPLC system (KNAUER Wissenschaftliche Geräte GmbH, Berlin, Germany) connected to a Bruker Maxis Impact mass spectrometer (Bruker Daltonics GmbH and Co. KG, Bremen, Germany), equipped with a Agilent Zorbax Eclipse C18 AAA column (4.6 × 150 mm^2^; 3.5 µm particles). Mobile phase A was water with 0.1% formic acid, and mobile phase B was acetonitrile with 0.1% formic acid. A gradient was set as follows: 0 min of 3% B, 3 min of 3% B, 11 min of 95% B, 20 min of 95% B, 22 min of 3% B and 30 min of 3% B. Injection volume was 10 µL, and flowrate was 1 mL/min. Dopamine, glutathione reduced (GSH), and glutathione oxidised (GSSG) were quantified according to *m/z* peaks at 154.091 ± 0.002, 308.089 ± 0.002, and 613.160 ± 0.002 Da, respectively. The detection of dopamine, GSH and GSSG was verified by analysis of their corresponding product/fragment ions in MS/MS mode: *m/z* 137.059 Da for dopamine, *m/z* 179.082 Da for GSH, and *m/z* 484.114 Da for GSSG. Each sample was injected and measured three times. The parameters of the mass spectrometer were the following: (i) ESI ion source parameters—End Plate Offset: 500 V, capillary: 4500 V, nebuliser (N_2_): 3.4 bar, dry gas (N_2_): 8.0 L/min, and dry temperature: 350 °C, (ii) Tune parameters—Funnel 1 RF: 250.0 Vpp, Funnel 2 RF: 350.0 Vpp, Hexapole RF: 250.0 Vpp, Quadrupole Ion Energy: 10.0 eV, Low Mass: 50.00 *m/z*; (iii) Collision Cell—collision energy: 10.0 eV, collision RF: 350.0 Vpp, transfer time: 100.0 µs, and pre-pulse storage: 10.0 µs. For MS/MS a multiple reaction monitoring mode was used with the following parameters: for all precursor ions: *m/z* 154.091, 308.089, and 613.160 Da, the isCID energy: 20.0 eV, and collision energy: 35 eV.

### 5.10. Data Presentation and Statistical Analyses

Data was analysed using Prism version 6.0 (GraphPad Software, San Diego, CA, USA). Bar graphs represent mean ± standard deviation (SD) and the numbers of replicates (*n*) is depicted in the figure legends. Statistical differences were evaluated by Student’s *t*-test (Figure 3, Figure 4 and Figure 8), and by one-way ANOVA with the Tukey’s post-test (Figure 3, Figure 4, Figure 6, Figure 7, Figure 9 and Figure 10) and with Dunnett’s post-test (Figure 7).

## Figures and Tables

**Figure 1 toxins-14-00011-f001:**
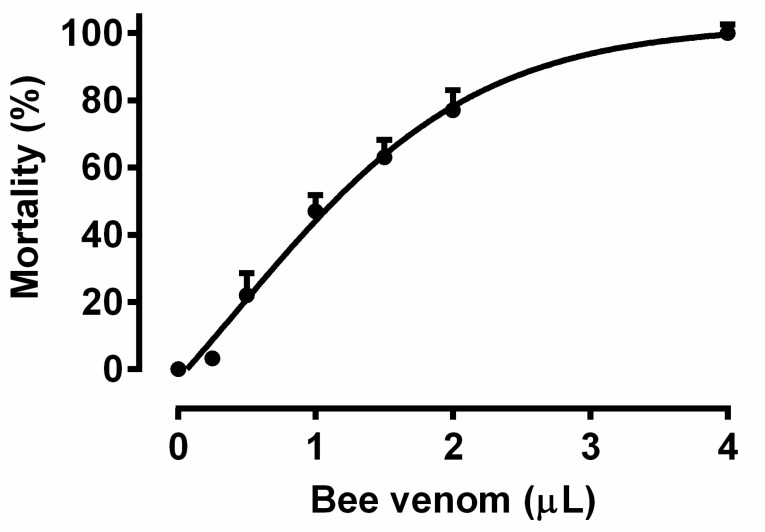
The effect of increasing doses of honeybee venom on mortality of *P. americana* adults 24 h after the treatment. Each point represents the mean ± SD; *n* = 3 groups with 15–20 individuals in each.

**Figure 2 toxins-14-00011-f002:**
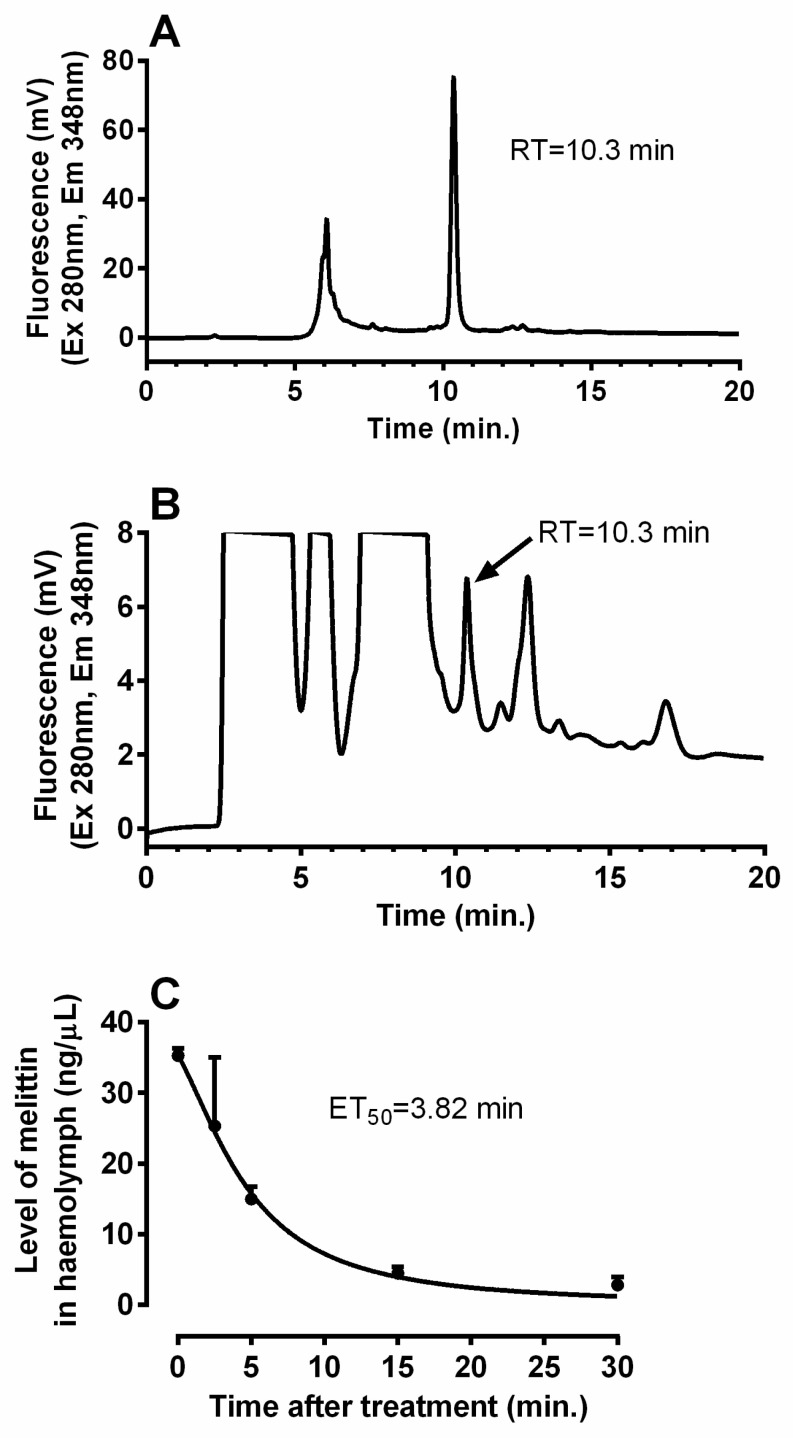
Temporal changes in the amount of melittin in the haemolymph of *P. americana* adults injected with 2 μL the honeybee venom. (**A**) Illustrative HPLC record of melittin standard (1 μg). (**B**) Illustrative HPLC record of haemolymph extract (5 μL) 5 min after venom treatment; no peak was observed in control haemolymph at RT = 10.3 min (data not shown). (**C**) Level of melittin in *P. americana* adult haemolymph at different times after the treatment. The half live of 3.82 min was estimated from plots of log melittin titre. Each point represents the mean ± SD (*n* = 3–4).

**Figure 3 toxins-14-00011-f003:**
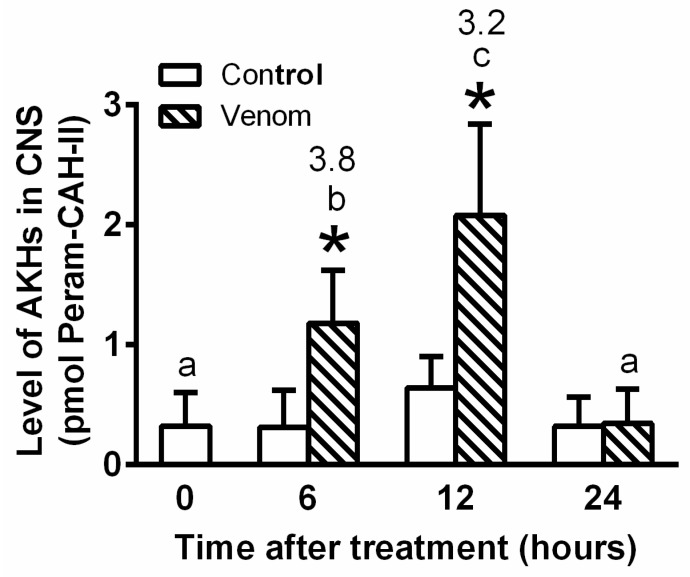
ELISA quantification of the Peram-CAH-I and Peram-CAH-II (together—AKH) in CNS of *P. americana* adults after the honeybee (0.5 μL) treatment; results are expressed in pmol of Peram-CAH-II (for explanation see Section 5.4). Statistically significant differences between the venom group and corresponding control (mean ± SD, *n* = 8) at each time-point evaluated by Student’s *t*-test on 5% level are indicated by * statistically significant differences among the time-points after the venom treatment evaluated by one-way ANOVA with Tukey’s post-test at the 5% level are indicated by different letters above the columns (a,b.c) (no differences were recorded among the controls). The numbers above the columns represent fold-difference of AKH level as compared with corresponding control.

**Figure 4 toxins-14-00011-f004:**
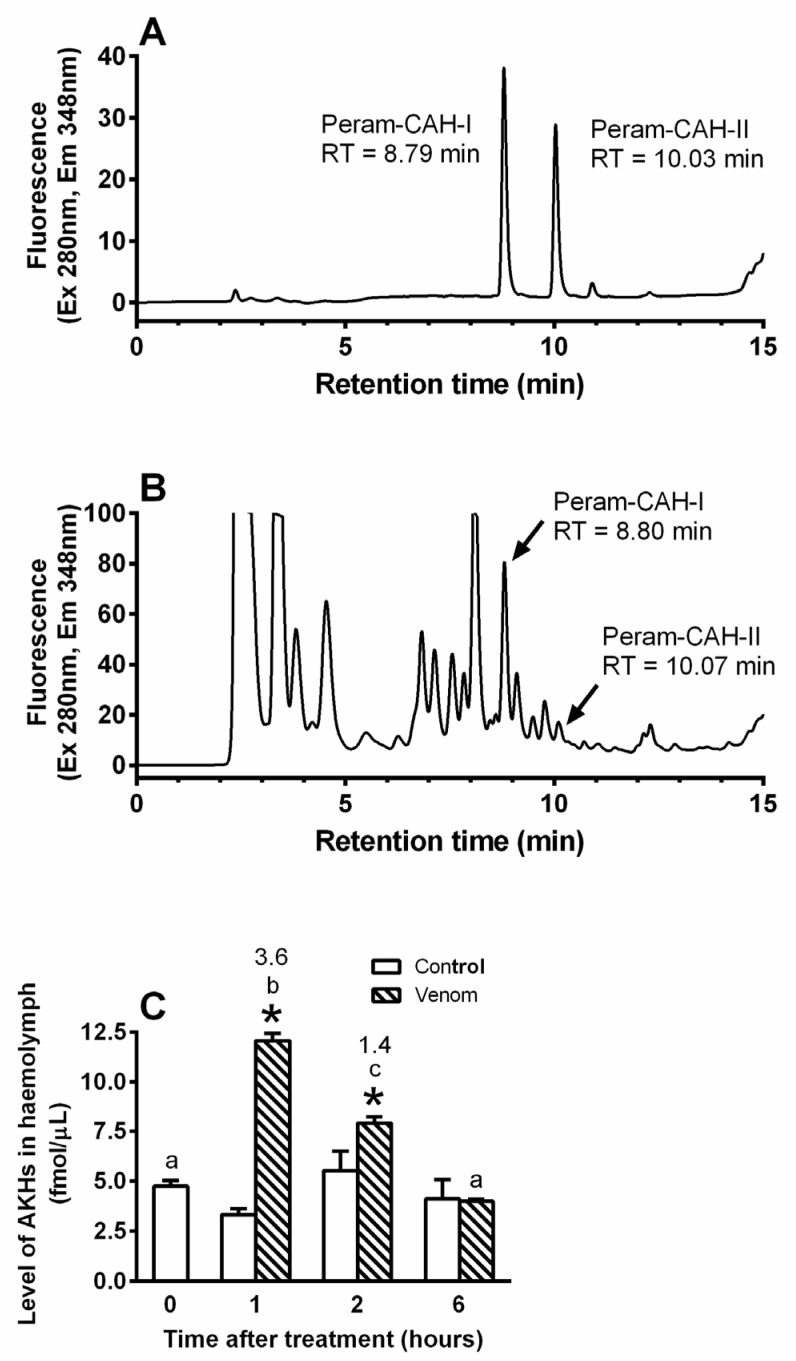
HPLC quantification of the Peram-CAH-I and Peram-CAH-II (together—AKHs) in haemolymph of *P. americana* adults after the honeybee (0.5 μL) treatment. (**A**) Illustrative HPLC record of AKH standards Peram-CAH-I and Peram–CAH-II (10 and 10 pmol, respectively). (**B**) Illustrative HPLC record of an extract from 100 uL of haemolymph (for details see Section 5.4). (**C**) Level of AKHs in *P. americana* adult haemolymph at different times after the treatment. Statistically significant differences between the venom group and corresponding control (mean ± SD, *n* = 3) at each time-point evaluated by Student’s *t*-test on 5% level are indicated by *; statistically significant differences among the time-points after the venom treatment evaluated by one-way ANOVA with Tukey’s post-test at the 5% level are indicated by different letters above the columns (a,b,c) (no differences were recorded among the controls). The numbers above the columns represent fold-difference of AKH level as compared with corresponding control.

**Figure 5 toxins-14-00011-f005:**
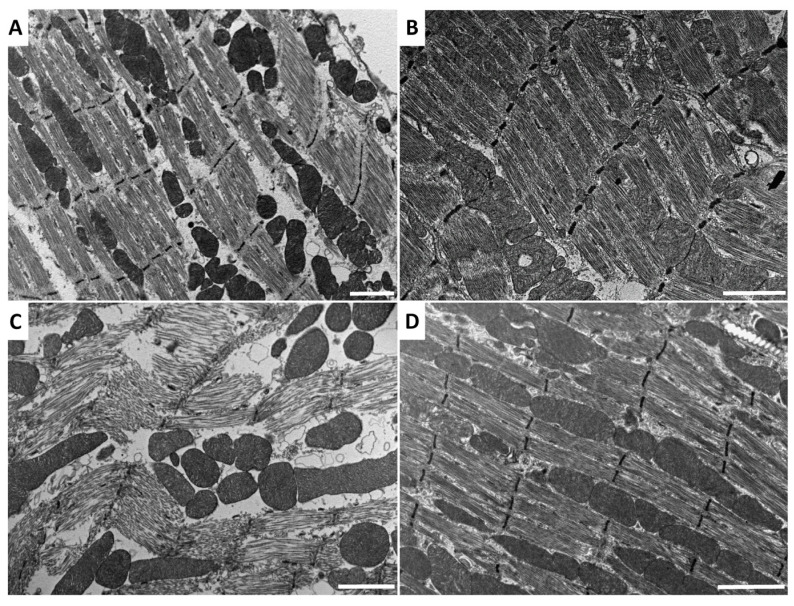
TEM photos of the thoracic muscles from *P. americana* adults 24 h after treatment with (**A**) Ringer saline (control), (**B**) 40 pmol Peram-CAH-II, (**C**) 0.5 uL bee venom, and (**D**) 0.5 uL bee venom +40 pmol Peram-CAH-II together. Scale bars in the Figs = 2 μm.

**Figure 6 toxins-14-00011-f006:**
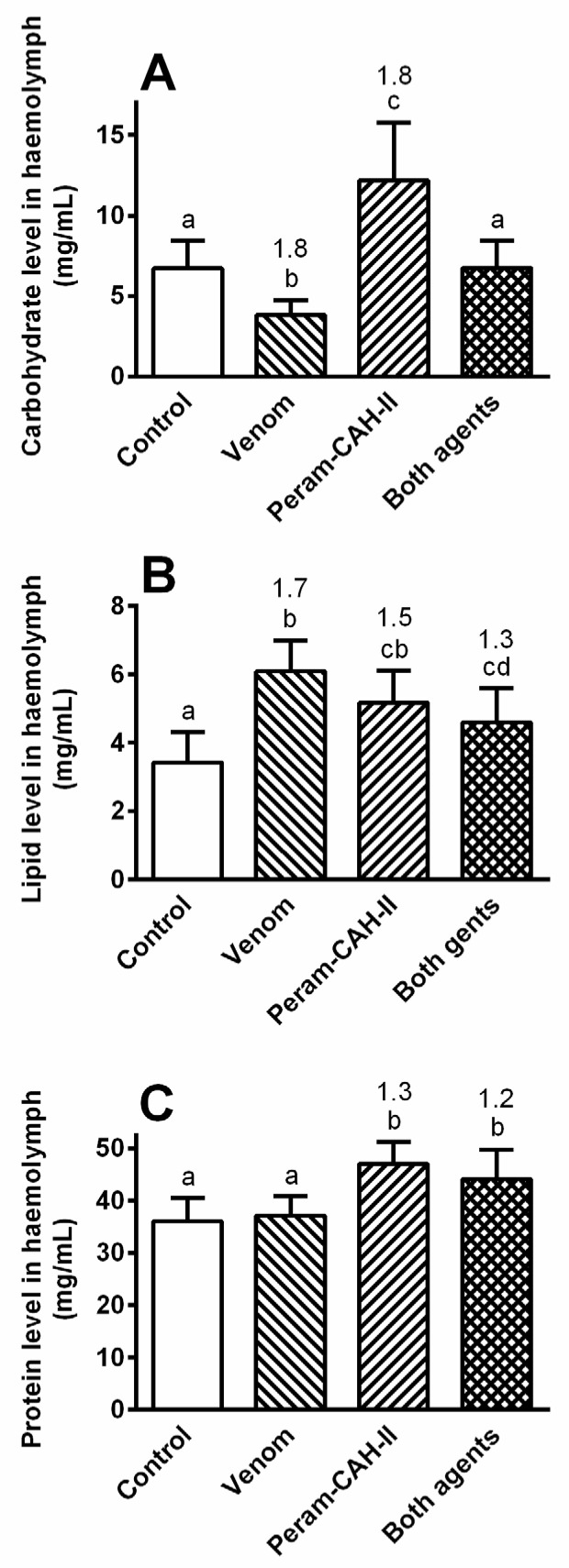
The effect of 0.5 μL honeybee venom and/or 40 pmol Peram-CAH-II on (**A**) carbohydrate, (**B**) lipid and (**C**) protein levels in the haemolymph of *P. americana* adults 6 h after the treatment. Statistically significant differences among the groups (mean ± SD) evaluated by one-way ANOVA with Tukey’s post-test at the 5% level are indicated by different letters above the columns (a,b,c) (*n* = 6–8). The numbers above the columns represent fold-difference of the particular nutrient level in treated cockroaches compared with that in controls.

**Figure 7 toxins-14-00011-f007:**
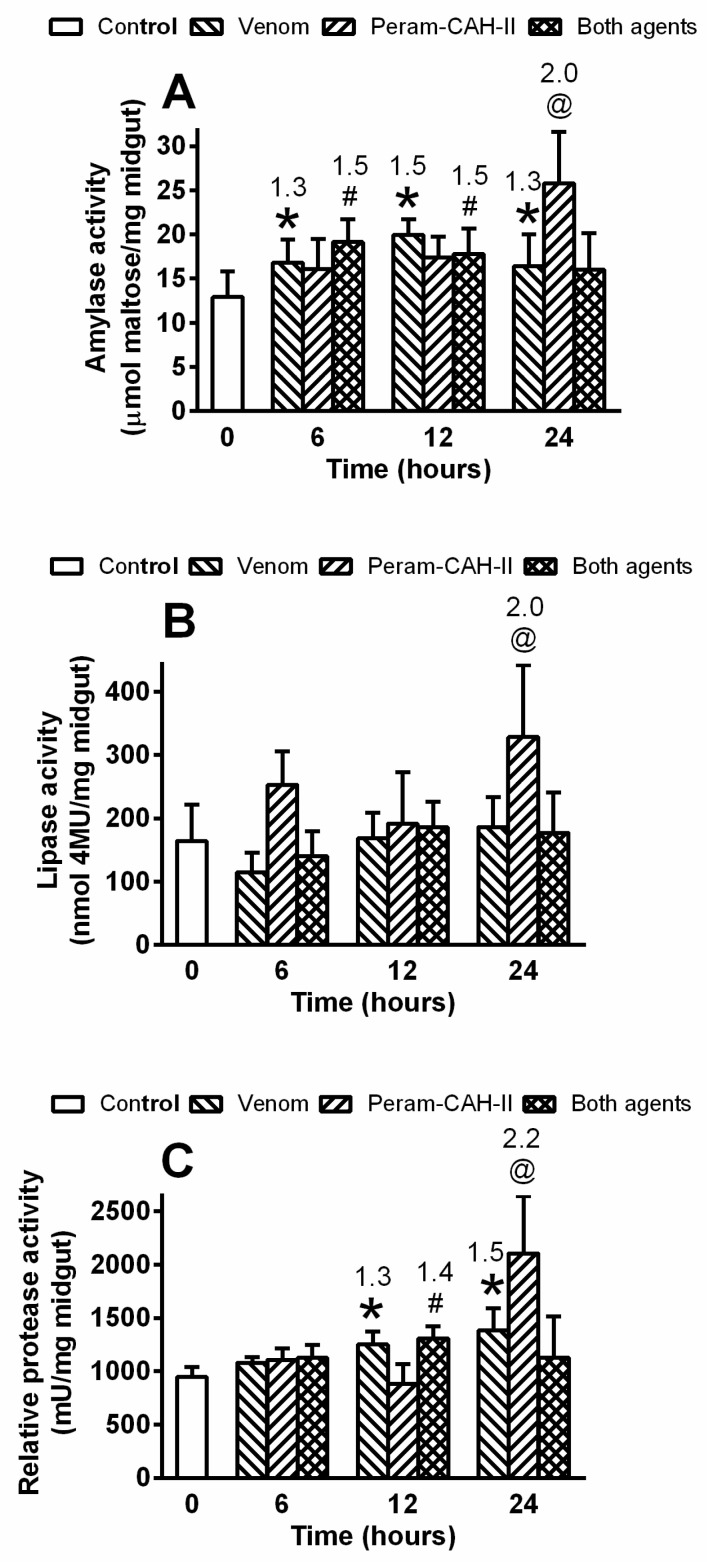
The temporal effect of 0.5 uL honeybee venom and/or 40 pmol Peram-CAH-II on (**A**) amylase, (**B**) lipase and (**C**) protease activities in the midgut of *P. americana* adults 6, 12 and 24 h after the treatment. Statistically significant differences between the control and venom groups, between the control and Peram-CAH-II groups, and between the control and both agents groups (all mean ± SD) evaluated by one-way ANOVA with Dunnett’s post-test at the 5% level are indicated by *, @ and #, respectively (*n* = 6–8). The numbers above the columns represent fold-difference of the particular enzyme activity in treated cockroaches compared with that in controls. For further statistics see the corresponding text.

**Figure 8 toxins-14-00011-f008:**
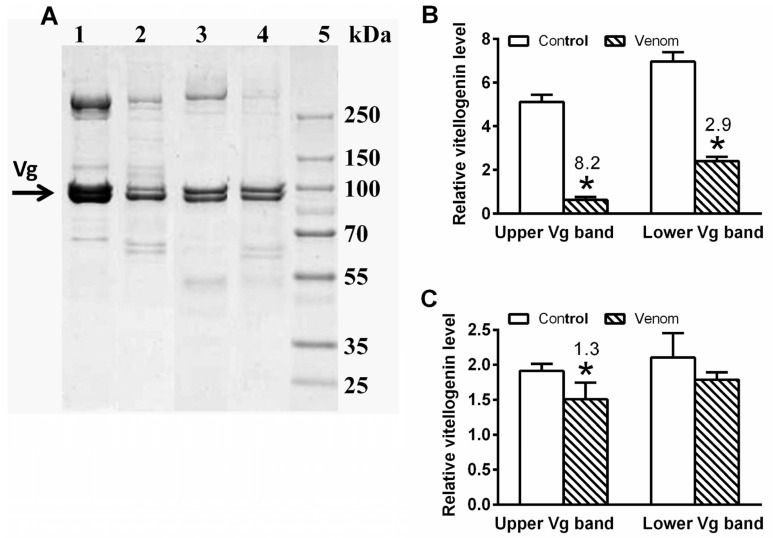
The effect of 0.5 uL honeybee venom on vitellogenin level in *P. americana* adult haemolymph 6 h after the treatment. (**A**)—illustrative 4–20% SDS polyacrylamide gel (females 0.4 uL and males 1 uL haemolymph/well: 1—female control, 2—female envenomed, 3—male control, 4—male envenomed; 5—molecular standards); (**B**)—vitellogenin quantification in females; (**C**)—vitellogenin quantification in males. Statistically significant differences between the venom groups and corresponding controls evaluated by Student’s *t*-test on 5% level are indicated by * (*n* = 6–8). The numbers above the columns represent fold-difference of the particular vitellogenin level in treated cockroaches compared with that in controls.

**Figure 9 toxins-14-00011-f009:**
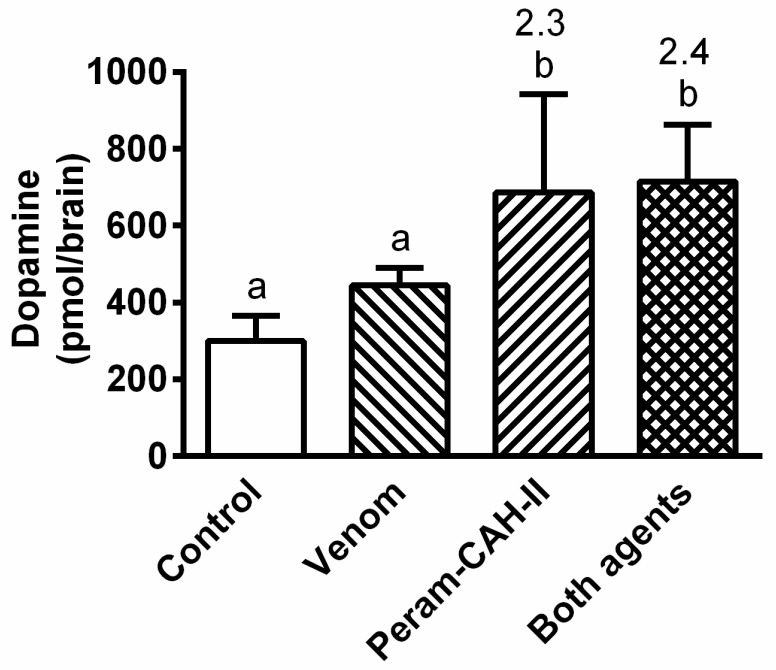
The effect of 0.5 uL honeybee venom and/or 40 pmol Peram-CAH-II on dopamine levels in the brain of *P. americana* adults 6 h after the treatment. Statistically significant differences among the groups (mean ± SD) evaluated by one-way ANOVA with Tukey’s post-test at the 5% level are indicated by different letters above the columns (a,b) (*n* = 6–8). The numbers above the columns represent fold-difference of the particular dopamine level in treated cockroaches compared with that in controls.

**Figure 10 toxins-14-00011-f010:**
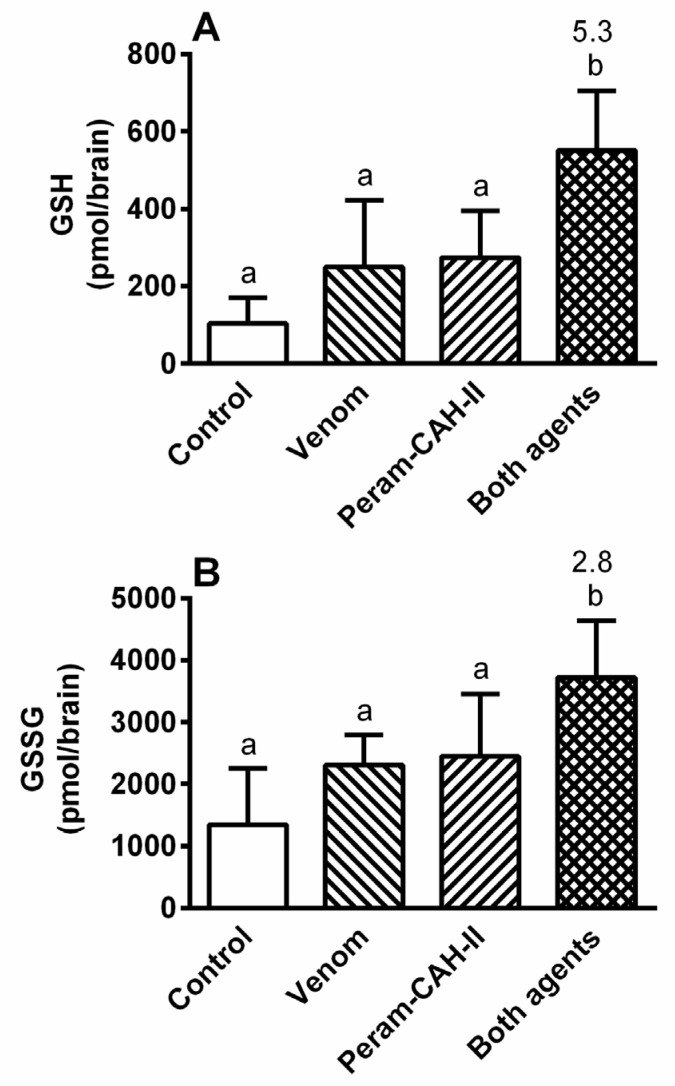
The effect of 0.5 uL honeybee venom and/or 40 pmol Peram-CAH-II on levels of (**A**) glutathione (GSH) and (**B**) its oxidised product (GSSH) in the brain of *P. americana* adults 6 h after the treatment. Statistically significant differences among the groups (mean ± SD) evaluated by one-way ANOVA with Tukey’s post-test at the 5% level are indicated by different letters above the columns (a,b) (*n* = 8). The numbers above the columns represent fold-difference of the particular glutathione levels in treated cockroaches compared with that in corresponding controls.

## Data Availability

Not applicable.

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
