# Peer review of "Insect Body Defence Reactions against Bee Venom: Do Adipokinetic Hormones Play a Role?"

_toxins, 2021, doi:10.3390/toxins14010011_

Round 1

Reviewer 1 Report

This review is interesting and important for understanding Insect body defence reactions against bee venom. I suggest some revisions in order to improve the manuscript.

(Comment 1) Regarding back matter, this journal required "Author contributions", "Funding", "IRB Statement", "Informed Consent Statement", "Data Availability Statement", and "Acknowledgment". I recommend authors to supplement this point at end of the manuscript. (pg 18, line 19-24)

- https://www.mdpi.com/journal/toxins/instructions

Introduction

(Comment 2) I recommend authors to supplement reference in end of the follow sentence.

- Bee venom intoxication elicits severe stress on the affected body, with the anti-stress response generally controlled by the nervous and endocrine systems [ref].

- In insects, anti-stress reactions are predominantly regulated by adipokinetic hormones (AKHs), which are responsible for maintaining homeostasis. AKHs are small peptides (8–10 amino acids in length) that are released from the corpora cardiaca, a small neuroendocrine gland connected to the brain [ref].

- This is a very popular model for physiological and biochemical studies and is predominantly used by American insect scientists [ref].

Discussion section

(comment 3) Bee venom is natural toxins. Thus, adverse events should be fully discussed. In order to reduce side effects and maximize the effectiveness, the appropriate purification method of crude bee venom, concentration and dose of bee venom usage are important.

- https://doi.org/10.3390/toxins13020105

Method section

(Comment 4) The composition of bee venom may differ depending on the area of the honeybee worker and the type of flowers distributed. Therefore, I recommend authors to supplement information about distribution area of worker bee and types of flowers.

Author Response

Thank you for your valuable comments and recommendations. We have explained and/or accepted all of them and improved the MS accordingly.

Reviewer 1:

This review is interesting and important for understanding Insect body defence reactions against bee venom. I suggest some revisions in order to improve the manuscript.

(Comment 1) Regarding back matter, this journal required "Author contributions", "Funding", "IRB Statement", "Informed Consent Statement", "Data Availability Statement", and "Acknowledgment". I recommend authors to supplement this point at end of the manuscript. (pg 18, line 19-24)

- https://www.mdpi.com/journal/toxins/instructions

Reply:

The text was edited according to the Instructions for Authors.

Introduction

(Comment 2) I recommend authors to supplement reference in end of the follow sentence.

- Bee venom intoxication elicits severe stress on the affected body, with the anti-stress response generally controlled by the nervous and endocrine systems [ref].

- In insects, anti-stress reactions are predominantly regulated by adipokinetic hormones (AKHs), which are responsible for maintaining homeostasis. AKHs are small peptides (8–10 amino acids in length) that are released from the corpora cardiaca, a small neuroendocrine gland connected to the brain [ref].

- This is a very popular model for physiological and biochemical studies and is predominantly used by American insect scientists [ref].

Reply:

First and second references were added into the text. The third point is not a scientific statement, it is only a well-known fact for which there is no reference.

 Discussion section

(comment 3) Bee venom is natural toxins. Thus, adverse events should be fully discussed. In order to reduce side effects and maximize the effectiveness, the appropriate purification method of crude bee venom, concentration and dose of bee venom usage are important.

- https://doi.org/10.3390/toxins13020105

Reply:

We are fully aware the facts you are mention. The bee venom is a complicated mixture of many toxins that work in cascades and feed-back loops. We are able to purify some toxins e.g. the main one – melittin. However, it is necessary to use the crude venom for physiological experiments to describe its general effect - using (all) individual toxins separately would be practically impossible, and in addition the results could perhaps be false negative. It is known that level of e.g. melittin fluctuates in the venom within the year (see e.g. our recent paper Kodrík et al., 2022, Comp. Biochem. Physiol. A 264: 111115), therefore we collected the venom from the June bees to have constant level of toxins in all our experiments. Further, it is not possible to differ direct and adverse effect after application of the venom into the cockroach body, because the corresponding literature is missing. Therefore the suggested discussion is irrelevant. On the other hand general toxicity of the bee venom is described in Introduction.

 Method section

(Comment 4) The composition of bee venom may differ depending on the area of the honeybee worker and the type of flowers distributed. Therefore, I recommend authors to supplement information about distribution area of worker bee and types of flowers.

Reply:

Fortunately, we have a list of plants that bloomed every month of the year in the vicinity of our apiary (originally prepared for other purposes). Thus, we added their list for June (when the venom for the experiments was collected) into the supplement (see also below). Further, we added the GPS coordinates into the MM text (48° 58′ 31.924″ N, 14° 26′ 44.671 ″E; 390 m)

June:

Robinia pseudoacacia, Trifolium pratense, Crataegus, Sinapis alba, Phacelia tanacetifolia, Fagopyrum esculentum, Linum usitatissimum, Papaver rhoeas, Matricaria chamomilla, Paeonia sp., Jasminum, Viburnum sp., Iris sp., Centaurea sp., Tilia sp., Leucanthemum sp.

Reviewer 2 Report

The manuscript “Insect body defence reactions against bee venom: do adipokinetic hormones play a role?” investigates the effect of honeybee venom on various physiological factors in American Cockroaches (Periplaneta americana) as well as whether adipokinetic hormones (AKH) alter these effects. The authors show that a moderate dose of bee venom (20% mortality) increases the presence of AKH in the cockroaches, and that while venom causes muscle cell damage, this damage is negated when AKH is administered alongside the venom. The authors also demonstrate that venom and AKH have variable effects on macromolecule (carbohydrates, lipids, proteins), vitellogenin, and dopamine levels and can alter midgut enzyme activity as well as oxidative stress in these cockroaches. The authors present their results in a clear and consistent manner and make an important contribution to our understanding of how insects defend against venoms.

Overall, I find this manuscript to be clearly written and without too much extraneous material. Most of writing in the manuscript is good as-is, however the methods section seems to have many small grammatical errors that are not found in the other sections of the manuscript. They don’t alter the reader’s ability to understand what was done, but this section should be looked over with the same care that the rest of the manuscript has received and these errors should be fixed. I find the author’s use of figures very clear and easy to follow, and the experimental design is easy to follow.

The introduction feels too short and it would benefit the manuscript if the authors expanded a bit on the rationale for why they chose to study these multiple processes in the cockroaches. For example, the authors discuss that AKH’s are part of insect anti-stress response and list a variety of processes that they are involved in, but then they jump right into introducing the cockroach as a model in the last paragraph of the introduction. I think that adding an additional paragraph (Page 2, between lines 12 and 13) would be very helpful to introduce the rationale for studying dopamine, gut enzymes, oxidative stress, etc. This way the reader has an understand of both what systems are being assessed, and why they are being assessed.

Most of my specific comments are focused on the discussion, as the results are largely clearly presented (though see comment above about the methods).

Throughout the discussion it would be helpful if the authors could refer to their figures when discussing each result (like on Page 14, Line 6). It makes it much easier to jump back to the figures to put the discussion in context.

Page 12 Lines 9-17: This paragraph makes some large claims that aren’t supported by any citations and would require very clear evidence to support them. Please either remove these parts or support them with citations and clarify the details (see specific notes below). Overall, I see that the point of this is to argue that we should expect there to be insect-specific effects in honeybee venom even though they are often defending against vertebrates, but I think you could remove most of this paragraph and still make that point or you could introduce other lines of evidence that honeybees need to defend against other insects.

Page 12 Lines 12-14: What is the evidence that stingers were used primarily against bees and other insects, and more specifically where is the evidence that vertebrates weren’t a significant factor at the time? Mammals (assuming this is what is meant by “soft-skinned vertebrates” certainly existed 120 million years ago, and I don’t know if anyone has investigated the interactions of bees and mammals at that time scale. More specifically, the phrase “properly developed on Earth” isn’t biologically accurate. It’s not clear what that is supposed to mean.

Page 12 Lines 16-17: What does it mean to say that the specific effects “apparently” developed? This implies that there is specific evidence for this, but none is presented. Similarly, this seems to imply that this was after the initial evolution of bee venom (but after wasps), but this assumes that the statement about vertebrates (line 13) is accurate. It’s possible that bee venom evolved concurrently to be effective against insects and vertebrates.

Page 12 Lines 21-22: I did a quick internet search and found this review (see below). In Table 1 they list the half life of melittin in various cells. While this isn’t technically in their bodies, it seems this would be worth reporting. Their half-lives are also much longer than found in this study (10s of hours), which may be worth discussing.

Melittin, a Potential Natural Toxin of Crude Bee Venom: Probable Future Arsenal in the Treatment of Diabetes Mellitus 2017

Md. Sakib Hossen, Siew Hua Gan , and Md. Ibrahim Khalil

https://doi.org/10.1155/2017/4035626

Page 13 Line 35: What does it mean for P. americana to “prefer” this metabolism. I briefly looked through the two citations but didn’t see an obvious answer to this. The authors should expand on this a little to make it clear what they mean, and perhaps using a different word choice would help as well (e.g. do they primarily use these carbohydrates? “prefer” implies a choice the cockroaches are making).

Author Response

Thank you for your valuable comments and recommendations. We have explained and/or accepted all of them and improved the MS accordingly.

Reviewer 2:

The manuscript “Insect body defence reactions against bee venom: do adipokinetic hormones play a role?” investigates the effect of honeybee venom on various physiological factors in American Cockroaches (Periplaneta americana) as well as whether adipokinetic hormones (AKH) alter these effects. The authors show that a moderate dose of bee venom (20% mortality) increases the presence of AKH in the cockroaches, and that while venom causes muscle cell damage, this damage is negated when AKH is administered alongside the venom. The authors also demonstrate that venom and AKH have variable effects on macromolecule (carbohydrates, lipids, proteins), vitellogenin, and dopamine levels and can alter midgut enzyme activity as well as oxidative stress in these cockroaches. The authors present their results in a clear and consistent manner and make an important contribution to our understanding of how insects defend against venoms.

 Overall, I find this manuscript to be clearly written and without too much extraneous material. Most of writing in the manuscript is good as-is, however the methods section seems to have many small grammatical errors that are not found in the other sections of the manuscript. They don’t alter the reader’s ability to understand what was done, but this section should be looked over with the same care that the rest of the manuscript has received and these errors should be fixed. I find the author’s use of figures very clear and easy to follow, and the experimental design is easy to follow.

Reply:

The text was edited by the Editage Author Services.

The introduction feels too short and it would benefit the manuscript if the authors expanded a bit on the rationale for why they chose to study these multiple processes in the cockroaches. For example, the authors discuss that AKH’s are part of insect anti-stress response and list a variety of processes that they are involved in, but then they jump right into introducing the cockroach as a model in the last paragraph of the introduction. I think that adding an additional paragraph (Page 2, between lines 12 and 13) would be very helpful to introduce the rationale for studying dopamine, gut enzymes, oxidative stress, etc. This way the reader has an understand of both what systems are being assessed, and why they are being assessed.

Reply:

Corresponding sentences were added in the text

Most of my specific comments are focused on the discussion, as the results are largely clearly presented (though see comment above about the methods).

Throughout the discussion it would be helpful if the authors could refer to their figures when discussing each result (like on Page 14, Line 6). It makes it much easier to jump back to the figures to put the discussion in context.

Reply:

Yes, it was added.

 Page 12 Lines 9-17: This paragraph makes some large claims that aren’t supported by any citations and would require very clear evidence to support them. Please either remove these parts or support them with citations and clarify the details (see specific notes below). Overall, I see that the point of this is to argue that we should expect there to be insect-specific effects in honeybee venom even though they are often defending against vertebrates, but I think you could remove most of this paragraph and still make that point or you could introduce other lines of evidence that honeybees need to defend against other insects.

Reply:

Thank you for you criticism and suggestions. Yes, the purpose of this paragraph was to explain that bee venom acts on both insects and mammals. The evolution of this phenomenon is clearly not our common topic, so we do not know the relevant literature well. Therefore, we have totally reworded this problematic paragraph.

 Page 12 Lines 12-14: What is the evidence that stingers were used primarily against bees and other insects, and more specifically where is the evidence that vertebrates weren’t a significant factor at the time? Mammals (assuming this is what is meant by “soft-skinned vertebrates” certainly existed 120 million years ago, and I don’t know if anyone has investigated the interactions of bees and mammals at that time scale. More specifically, the phrase “properly developed on Earth” isn’t biologically accurate. It’s not clear what that is supposed to mean.

Reply:

See above

Page 12 Lines 16-17: What does it mean to say that the specific effects “apparently” developed? This implies that there is specific evidence for this, but none is presented. Similarly, this seems to imply that this was after the initial evolution of bee venom (but after wasps), but this assumes that the statement about vertebrates (line 13) is accurate. It’s possible that bee venom evolved concurrently to be effective against insects and vertebrates.

Reply:

See above

 Page 12 Lines 21-22: I did a quick internet search and found this review (see below). In Table 1 they list the half life of melittin in various cells. While this isn’t technically in their bodies, it seems this would be worth reporting. Their half-lives are also much longer than found in this study (10s of hours), which may be worth discussing.

 Melittin, a Potential Natural Toxin of Crude Bee Venom: Probable Future Arsenal in the Treatment of Diabetes Mellitus 2017

Md. Sakib Hossen, Siew Hua Gan , and Md. Ibrahim Khalil

https://doi.org/10.1155/2017/4035626

Reply:

We made an intensive effort to find any data on the half-life of melittin, but we did not find any relevant data. Thus thank you for your recommendation. On the other hand the results from the Table 1 of the paper are rather unreliable. There is no reference (if the data come from literature), and there is no information, how the data were obtained (it they were obtained by the authors). We are also convinced that in a living body with an active defense system, the half-life is much shorter. Therefore we would like to avoid citation of this problematic paper.

Reviewer 3 Report

The manuscript describes the effect of bee venom on the levels of adipokinetic hormones, haemolymph nutrients, vitellogenins, dopamine, and oxidative stress markers; the activity of digestive enzymes; and the muscle ultrastructure in the body of the model species, the Periplaneta americana, but the work involved in this study seems to be too preliminary. It will be acceptable if further work can be conducted and in-depth explanation/conclusion can be provided.

1. The significance and novelty of this study and its contribution to the field should be mentioned more clearly.

2. Subtitles are needed in Results section.

3. The relationship between venom and Peram-CAH-II seems unclear. Based on the current results, it cannot draw the conclusion that some of the effects caused by bee venom are modulated by Peram-CAH-II, unless additional experiments are conducted, such as gene knockdown.

4. This work evaluates some physiological levels in Periplaneta americana treated with bee venom. It would be interesting to further investigate the correlation between behavioral /physiological changes and these markers.

Author Response

Thank you for your valuable comments and recommendations. We have explained and/or accepted all of them and improved the MS accordingly.

Reviewer 3:

The manuscript describes the effect of bee venom on the levels of adipokinetic hormones, haemolymph nutrients, vitellogenins, dopamine, and oxidative stress markers; the activity of digestive enzymes; and the muscle ultrastructure in the body of the model species, the Periplaneta americana, but the work involved in this study seems to be too preliminary. It will be acceptable if further work can be conducted and in-depth explanation/conclusion can be provided.

  1. The significance and novelty of this study and its contribution to the field should be mentioned more clearly.

Reply:

The novelty of the obtained results was emphasized in Conclusions.

  1. Subtitles are needed in Results section.

Reply:

Subtitles were added into Results as requested

  1. The relationship between venom and Peram-CAH-II seems unclear. Based on the current results, it cannot draw the conclusion that some of the effects caused by bee venom are modulated by Peram-CAH-II, unless additional experiments are conducted, such as gene knockdown.

Reply:

As bee venom contains a number of different toxins, it is not easy (possible?) to simply describe the mechanism of the relationship between the venom and AKH, although some individual mechanisms are described for both venom and AKH. This is definitely a multi-year job that goes beyond the scope of this paper. The main goal of this study was to show that at least some toxic effects of the bee venom are eliminated/modulated by AKH in cockroach body. We believe that our approach to test effect of (1) venom, (2) AKH, (3) venom+AKH and (4) control for several markers obviously showed mutual interactions.

  1. This work evaluates some physiological levels in Periplaneta americana treated with bee venom. It would be interesting to further investigate the correlation between behavioral /physiological changes and these markers.

Reply.

Yes, behavioural aspects would be interesting, however, we have no suitable facilities for the behavioural experiments.

Round 2

Reviewer 3 Report

The authors had addressed most of the proposed suggestions. The present version of MS is suitable to be published.